# Effects of *Salmonella* Typhimurium Infection on the Gut Microbiota of Cherry Valley Meat Ducks

**DOI:** 10.3390/microorganisms12030602

**Published:** 2024-03-18

**Authors:** Yue Zheng, Xue Pan, Jialei Hou, Wenchong Shi, Shuhong Sun, Mengze Song, Zheng Gao

**Affiliations:** 1State Key Laboratory of Crop Biology, Shandong Agricultural University, Tai’an 271018, China; 2022010158@sdau.edu.cn (Y.Z.); shiwenchong@sdau.edu.cn (W.S.); 2College of Life Sciences, Shandong Agricultural University, Tai’an 271018, China; 3Shandong Provincial Key Laboratory of Animal Biotechnology and Disease Control and Prevention, Shandong Agricultural University, Tai’an 271018, China; 2023110436@sdau.edu.cn (X.P.); 2023120610@sdau.edu.cn (J.H.); sunshuhong@sdau.edu.cn (S.S.); 4Department of Preventive Veterinary Medicine, College of Veterinary Medicine, Shandong Agricultural University, Tai’an 271018, China

**Keywords:** *Salmonella* Typhimurium, gut microbiota dysbiosis, ducks, *Ruminococcaceae*, 16S rRNA

## Abstract

*Salmonella* infection causes serious economic losses, threatens food safety, and is one of the most important diseases threatening meat duck farming. The gut microbiome is critical in providing resistance against colonization by exogenous microorganisms. Studying the relationship between *Salmonella* and gut microbiota can help us better understand the threat of the pathogenic mechanism of *Salmonella* and provide a more scientific theoretical basis for its prevention and treatment. This study uses *Salmonella* Typhimurium as the research object and Cherry Valley meat duck as the model with which to study the impact of *Salmonella* infection on ducks. In this field trial, 2 × 10^8^ CFUs *Salmonella* Typhimurium were administered to 3-day-old ducks. After infection, duck viscera were collected to detect the colonization of *Salmonella*, and cecal contents were collected to analyze the changes in gut microbiota. The results show that *Salmonella* Typhimurium can colonize ducks three days after infection and alter the gut microbiota composition, mainly by increasing the abundance of *Ruminococcaceae* and *Lachnospiraceae*. In conclusion, *Salmonella* Typhimurium infection significantly alters the intestinal microbiota of ducks and poses a serious public health risk.

## 1. Introduction

Duck is an important waterfowl that is widely raised in China, and duck meat has been a very popular traditional food for a long time. At present, there are many large-scale farming bases in China. Gastrointestinal diseases are important factors limiting the development of poultry in these large-scale commercial farms. *Salmonella* infection is one of the important diseases that threatens meat duck farming, causing serious economic losses and threatening food safety [1]. Duck salmonellosis can spread vertically and horizontally and mainly causes death in ducklings [2]. Poultry products contaminated with *Salmonella* can seriously impact the life, health, and safety of consumers. After *Salmonella* enters the intestine through the digestive tract, it first competes with the gut microbiota and then penetrates the mucus layer on the intestinal surface through flagella and pili and adheres to the intestinal surface [3,4]. After attaching to the intestinal epithelium, *Salmonella* usually expresses T3SS, causing intestinal inflammation and creating an intestinal environment suitable for *Salmonella* proliferation and spread [5,6]. In recent years, the problem of bacterial resistance caused by the abuse of antibiotics has made the prevention and treatment of *Salmonella* more difficult. Gut microbial therapy is a promising alternative to antibiotics. Microbial communities play a fundamental role in regulating immunity in the gastrointestinal tract [7]. In healthy individuals, the intestinal microbiota provides protection against infection through multiple mechanisms, including production of antimicrobial products, competition for nutrients, maintenance of epithelial barriers, phagocytosis, and modulation of the immune system [8].

This study used the *Salmonella* Typhimurium strain isolated from sick ducks and employed Cherry Valley meat ducks as a model with which to explore the impact of infection with *Salmonella* Typhimurium on meat ducks. By analyzing changes in intestinal flora to find a class of microorganisms that can prevent or treat *Salmonella* infections. This research helps us better understand the pathogenic mechanism of *Salmonella* and helps solve the problem of *Salmonella* infection in meat duck farming.

## 2. Materials and Methods

### 2.1. Experimental Design of the Duck Model

All animal experiments were carried out in strict accordance with animal protocols approved by the Ethical Committee of Animal Experiments of Shandong Agricultural University (permit number SDUAA-2021-036). The Cherry Valley meat ducks used in the experiment were divided into two groups, with 30 chicks per group, and each group was housed in a separate air-filtered isolation cabinet in different rooms where they were provided feed and water ad libitum. The chicks in the SA group (*Salmonella* Typhimurium challenged) were challenged orally at day 3 with a single dose of 2 × 10^8^ CFUs *Salmonella* Typhimurium in a total volume of 0.5 mL. Subjects in the control group were orally inoculated with the same volume of bacteria-free, phosphate-buffered saline (PBS) (Beijing Solarbio Science & Technology Co., Ltd., Beijing, China). Chicks were maintained at an age-appropriate temperature for the length of the experimental period (9 days).

### 2.2. Bacterial Strains and Growth Conditions

*Salmonella* Typhimurium 20JS04 strain was isolated from sick ducks in a Cherry Valley meat duck factory in Shandong Province, China in 2022. *Salmonella* strains stored at −80 °C were first streaked onto a xylose–lysine desoxycholate (XLD) (Qingdao Hope Bio-Technology Co., Ltd, Qingdao, China) agar plate and incubated at 37 °C for 24 h for subsequent tests. Before infection, single colonies were picked and cultured in 5 mL Lysogeny broth (Qingdao Hope Bio-Technology Co., Ltd., Qingdao, China) with shaking for 16 h at 37 °C. The bacteria were collected by centrifugation, resuspended in physiological saline, counted, and the Salmonella was mixed into 4 × 10^8^ CFU/mL using physiological saline, with each duck administered 0.5 mL.

### 2.3. Sample Collection and Processing

At 3- and 6-days post infection (dpi), 8 ducks were randomly selected from each group and sacrificed to collect internal organs under sterile conditions. Sample of heart, liver, spleen, thymus, and bursa organ index were recorded (n = 8). Meanwhile, the heart, liver, spleen, thymus, and bursa were additionally collected from same sample ducks to detect the number of *Salmonella* bacteria (n = 6). The liver, spleen, and cecum were collected in multiple cryogenic tubes and placed in a liquid nitrogen tank and then preserved at −80 °C. The contents of the cecum were later used to perform the 16S rRNA amplicon sequencing and bioinformatic analyses of the gut microbiota (n = 8). At the same time, six of these samples were selected for quantitative real-time PCR (QPCR) (n = 6).

### 2.4. Bacterial Burden

Samples of the heart, liver, spleen, thymus, and bursa were taken to determine the number of *Salmonella* bacteria. The tissue samples were weighed and homogenized in PBS, and serial dilutions of the homogenates were plated onto XLD plates (37 °C, 16 h) for the counting of bacteria.

### 2.5. Real-Time PCR

For ducks RNA isolation, liver, spleen, and cecum were homogenized in a tissue homogenizer (BioSpec Products, Bartlesville, OK, USA), and RNA was isolated by TRIzol Reagent (Thermo Fisher Scientific, Waltham, MA, USA), following the manufacturer’s protocol, and then stored at −80 °C for qPCR, with *β-actin* used as an internal reference. qPCR was performed using SYBR Green Master Mix (Thermo Fisher), PCR mix, and the appropriate primer sets (Table 1). qPCR reactions were run in a 10 μL reaction mixture using an ABI 7500 Detection System (Applied Biosystems, Carlsbad, CA, USA). The RNA was solubilized in RNase-free water. RNA quantity and quality were evaluated using a NanoDropTM 2000 spectrophotometer (Thermo Fisher Scientific, Waltham, MA, USA), followed by cDNA synthesis via the Transcriptor First-Strand cDNA Synthesis Kit (Roche, Basel, CH) using 2 µg RNA template. The PCR procedure involved maintaining a temperature of 95 °C for 30 s, 40 cycles of 95 °C for 5 s, and then 60 °C for 30 s.

### 2.6. Bioinformatic Processing and Analysis of the Sequencing Data

Total DNA was extracted with DNAiso Plus Reagent (TransGen Biotech, China). To obtain the V3-V4 region of the 16S rRNA gene, the corresponding fragment was amplified using the primers 338F (5′-ACTCCTACGGGAGGCAGCA-3′) and 806R (5′-GGACTACHVGGGTWTCTAAT-3′). The purified amplicons were sequenced on the Illumina MiSeq platform (BGI Genomics Co., Ltd., China). A total of 4,631,939 16S rRNA raw reads were obtained from the sequenced samples, and 7432 amplicon sequence variant (ASV) strains were identified. QIIME2 (http://qiime.org/index-qiime2.html, accessed on 16 November 2023) was used to assemble quality-filtered reads into amplicon sequence variants using DADA2 (v. 1.1). Taxonomy assignment was performed based on the SILVA 138 database. Alpha- and beta-diversity indices were calculated based on the rarefied ASV table at a depth of 24,112 sequences per sample. All subsequent analyses in this study were based on the rarefied data. The ACE, Chao1, Gini Simpson, Pielou, Richness, and Shannon indices were calculated using the diversity function of the “vegan” (v. 2.4-4) package from R. The difference of alpha-diversity indices between groups was performed using the Wilcoxon test. The Bray–Curtis metric was calculated using the vegdist function from the R package “vegan”. Principal coordinates analysis (PCoA) and nonmetric multidimensional scaling (NMDS) were performed using the pcoa and metaMDS functions from the R package “vegan” with the Bray–Curtis metric. The adonis and anosim tests were calculated using R package “vegan”. The DESeq2 was calculated using R package “DESeq2”. To screen for significantly different strains, only strains with a total count of a single ASV greater than 1000 reads were selected for screening data. The screening data were used to calculate significantly different strains between the two groups using the DESeq2 method. Linear discriminant analysis effect size (LEfSe) from the Galaxy/Hutlab online (http://huttenhower.sph.harvard.edu/galaxy/, accessed on 16 November 2023) was used to conduct linear discriminant analysis (LDA > 2.0, *p* < 0.05). DESeq2 was run with default thresholds and standard procedures in order to detect differentially enriched taxa at the ASV level.

### 2.7. Metabolic Pathways Prediction

Based on 16S rRNA sequencing reads, PICRUSt [9] was used to evaluate the functional and metabolism of the duck feces microbiota, and to further analyze the metabolic pathways from all of the domains of life (MetaCyc) in order to predict potential functions (https://metacyc.org/, accessed on 18 November 2023).

### 2.8. Statistical Analysis

The data are expressed as means ± SD. The results were analyzed using one-way ANOVA in the Statistical Analysis Systems statistical software package (Version 8e; SAS Institute Inc., Cary, NC, USA). Differences between means were evaluated using Duncan’s significant difference tests. Means were considered significant at *p* < 0.05.

## 3. Results

### 3.1. Growth Performance after Infection

As shown in Figure 1, SA had not affected the body weight of the ducks on 3 and 6 dpi (Figure 1A). Additionally, it did not affect the index of the heart, liver, spleen, thymus, and bursa on 3 dpi. However, it significantly increased the index of the heart, liver, spleen, thymus, and decreased the bursa index on 6 dpi (*p* < 0.05) (Figure 1B–F).

### 3.2. Salmonella Typhimurium Translocation

As shown in Figure 2, by measuring the bacterial load of *Salmonella* Typhimurium in the heart, liver, spleen, thymus, and bursa on 3 and 6 dpi, we observed that the *Salmonella* Typhimurium colonization was observed in various organs. The *Salmonella* Typhimurium load in the heart, liver, spleen, and thymus on 6 dpi was higher than that on 3 dpi, while the bursa was lower. Interestingly, this appears to be consistent with changes in the organ index.

### 3.3. Inflammatory Cytokine Expression

The relative expressions of inflammatory cytokines in the liver and spleen at 6 dpi were examined. There was no difference in the relative expression of *TNF-α* and *NF-κB* mRNA in the liver between the *Salmonella* Typhimurium and the control group at 6 dpi (Figure 3A). The relative expression of *TNF-α* (*p* < 0.05) and *NF-κB* (*p* < 0.05) mRNA in the spleen of the *Salmonella* Typhimurium group was significantly higher than the control (Figure 3B).

### 3.4. Diversity Analysis of Gut Microbiota after Salmonella Infection

Diversity indices, such as Chao1, ACE, Simpson, and Shannon, are important indicators for evaluating the homeostasis of gut microbiota and can reflect the abundance and diversity of microbial communities [10]. Although the abundance of the control group was higher than the SA on 3 dpi, there was no significant difference after calculation (Figure 4A). Results on 6 dpi also showed no significant differences between the two groups (Figure 4B). Principal coordinate analysis (PCoA) (Figure 4C,D) and nonmetric multidimensional scaling (NMDS) (Figure 4E,F) were attempted to verify group differences. Both PCoA and NMDS showed a significant separation between control and SA, indicating a significant difference in the structure of gut microbes them on 3 dpi (Adonis, R^2^ = 0.119, *p* = 0.035; Anosim, R = 0.186, *p* = 0.036) and 6 dpi (Adonis, R^2^ = 0.142, *p* = 0.003; Anosim, R = 0.248, *p* = 0.003) (Figure 4C–F).

### 3.5. Compositional Change in Gut Microbiota Induced by Salmonella

Taxonomic tree plots were used to represent the taxonomic information of the two groups at each level. The gut microbiota of ducklings mainly consists of *Proteobacteria* and *Firmicutes*. *Salmonella* infection does not significantly impact the phylum level gut microbiota (Figure 5A,C). At the genus level, the abundance of *unclassified_Lachnospiraceae*, *Oscillospira*, *Runminococcus*, and *unclassified_Ruminococcaceae* increased in the SA group on 3 dpi; the abundance of *[Ruminococcus]*, *Enterococcus*, and *Butyricicoccus* were decreased. On 6 dpi, the abundance of *unclassified_Lachnospiraceae*, *Oscillospira* and *Ruminococcus* increased in the SA group, while the abundance of *unclassified_Ruminococcaceae*, *Enterococcus*, and *Butyricicoccus* decreased (Figure 5B,D).

To further investigate the differential gut microbes of the SA and control groups, the LEfSe method was applied to identify differences in taxonomic abundance between them. The different microbes with LDA scores > 2 and *p* < 0.05 were selected as differential microbes. The results show that the abundances of *Ruminococcaceae*, *unclassified_Ruminococcaceae*, and *Oscillospira* in the SA group were significantly higher than in the control on 3 dpi. The abundance of *Ruminococcus* in the SA group was higher than in the control group on 6 dpi, though *Butyricicoccus* was lower (LDA score > 4) (Figure 6A–D).

The differential bacterial strains of different comparisons were visualized using volcano plots (Figure 7A,B). We used the DESeq2 method to calculate the bacterial strains in the two groups. At 3 dpi, 55 of 7432 strains were screened, of which 5 were up-regulated and 6 down-regulated. At 6 dpi, 60 of 7432 differential strains were screened, of which 6 were up-regulated and 10 down-regulated. At 3 dpi, the most significant differences were the up-regulated ASV_6282, ASV_3867, ASV_10517, ASV_9097, and ASV_3653, and the down-regulated ASV_7943, ASV_2867, and ASV_8936. At 6 dpi, the most significant differences were the up-regulated ASV_464, ASV_5638, ASV_10517, and ASV_6430, and the down-regulated ASV_3796, ASV_10443, ASV_6554, and ASV_4472. The impact of *Salmonella* on the gut microbiota at the family level changed the abundance of *Ruminococcaceae* and *Lachnospiraceae*. At the genus level, the abundance of *Ruminococcus* and *Oscillospira* was enriched in the SA group, while the *Enterococcus* and *Weissella* were depleted (Table 2).

### 3.6. Metabolic Pathway Analysis of Potential Metabolite

The metabolic pathways were predicted by the PICTUSt software packages, depending on the 16S rRNA sequences. The results were compared with the MetaCyc database to obtain potential metabolite information (https://metacyc.org/, accessed on 18 November 2023). This study used the DESeq2 method to calculate the key different metabolic pathways in the two groups. The results show that a total of 396 metabolic pathways were detected. Five pathways were depleted at 3 dpi and one pathway was down-regulated at 6 dpi after *Salmonella* infection. Peptidoglycan biosynthesis II (PWY-5265) showed simultaneous depletion in both time points. Thirteen pathways were enriched on 3 dpi, and eleven pathways on 6 dpi. Among these, pyruvate fermentation to butanoate (CENTFERM-PWY), L-lysine fermentation to acetate and butanoate (P163-PWY), glutaryl-CoA degradation (PWY-5177), acetyl-CoA fermentation to butanoate II (PWY-5676), succinate fermentation to butanoate (PWY-5677), superpathway of *Clostridium acetobutylicum* acidogenic fermentation (PWY-6590), L-1,2-propanediol degradation (PWY-7013), and TCA cycle VIII (REDCITCYC) were all enriched on 3 dpi and 6 dpi (Figure 8).

## 4. Discussion

*Salmonella* is the most important foodborne pathogen in poultry production systems and can infect humans through consumption of contaminated food [11]. The World Health Organization (WHO) estimates that diarrhea caused by *Salmonella* kills hundreds of thousands of people every year [12]. Ducks are widely raised in China and are an important vector for the spread of *Salmonella* through the food supply chain. Non-typhoidal *Salmonella*e, such as *Salmonella* Typhimurium, are most often acquired through a food source [13]. We simulated a *Salmonella* Typhimurium infection by oral administration. *Salmonella* contact with the intestinal epithelium activates the type III protein secretion system, which stimulates host cell responses and ultimately leads to inflammation. An inflamed gut helps create a new ecological niche that is conducive to *Salmonella* growth. This study has shown that the body weight of ducks was not significantly affected within six days of *Salmonella* infection, but that the organ index changed significantly after infection, and that the presence of *Salmonella* could be detected in the organs on 3 dpi. Up-regulated expression of inflammatory cytokines following infection was also detected. This confirms the success of the model’s construction and proves that *Salmonella* poses a significant threat to ducks and public health. It has been known for years that the gut microbiome protects against *Salmonella* infections [14]. Recent studies have shown that microbiota compete with invading pathogens for nutrients and are important for maintaining intestinal homeostasis [15,16]. However, there is a lack of knowledge regarding the impact of *Salmonella* infection on the gut microbiota of ducks. Therefore, the objective of this study was to investigate the effects of *Salmonella* infection on gut microbiota homeostasis in ducks.

A balanced gut microbiota contributes to health, and a mature microbiota provides protection against opportunistic pathogenic infections by preventing the engraftment of new microorganisms through competition and habitat filtration. However, at high infectious doses, the pathogen can overcome colonization resistance by using its virulence factors to induce intestinal inflammation [17]. Previous research has shown that *Salmonella* infection can significantly impact the diversity of chicken gut microbiota [18,19]. Nevertheless, there is almost no research on the impact of *Salmonella* on duck intestinal microbial diversity. This study observed that *Salmonella* infection had a limited impact on the α-diversity of ducks; however, PCoA and NMDS analyses found significant changes in the composition of gut microbiota in the SA group. The results indicate that *Firmicutes* and *Proteobacteria* were the most predominant bacterial phyla in the control and SA groups. Although the types of major bacterial phyla did not change, the abundance of *Firmicutes* increased and the abundance of *Proteobacteria* decreased in the SA group after *Salmonella* infection at 3 dpi. Notably, some bacterial genera with decreased or increased numbers, such as *Ruminococcus* and *Oscillospira*, increased, and *Enterococcus*, *Weissella* and *Butyricicoccus* decreased after infection. Similar results obtained by different analyses indicate that *Ruminococcaceae* and *Lachnospiraceae* may be the gut microbiota that are most closely related to *Salmonella* infection. *Ruminococcaceae* and *Lachnospiraceae* are the most abundant *Firmicute* families in gut environments, share a common role as active plant degraders, and have been implicated in the maintenance of intestinal health [20,21]. Gut microbiota can control harmful microbial invasion through habitat filters, including diet, host-derived resources, and microbiota-derived metabolites such as short-chain fatty acids [17]. Many studies have shown that butyric acid, one of the short-chain fatty acids, has the effect of inhibiting *Salmonella* [22,23,24]. *Ruminococcaceae* [25,26] and *Lachnospiraceae* [27] are important producers of butyric acid in the intestine. Studies have reported that the abundance of *Ruminococcaceae* and *Lachnospiraceae* in the intestinal tract of chicks infected with *Salmonella* is inversely proportional to the abundance of *Enterobacteriaceae* [18]. However, this study found that the number of *Ruminococcaceae* increased at 3 dpi and 6 dpi after *Salmonella* Typhimurium infected meat ducks. Results from analyses predicting gut microbiota function also found significant enrichment in metabolic pathways related to butyrate production at two time points following *Salmonella* Typhimurium infection. There may be various reasons for this result, such as the difference in the composition of the gut microbiota of ducks and chickens, and the effects of diverse *Salmonella* species on the gut microbiota may also be different. The host and host intestinal microbiota may have begun to return to homeostasis by 3 dpi, possibly as a result of the challenge dose and sampling time point. *Ruminococcaceae* is also a complex collection of bacteria, with many bacteria having different functions. For example, *Ruminococcus* can degrade and convert complex polysaccharides into a variety of nutrients for the host [28]. However, with the advancement of sequencing technology, studies have shown that species within the genus *Ruminococcus* belong to two bacterial families: *Ruminococcaceae* and *Lachnospiraceae* [29]. Some of these bacteria, such as *Ruminococcus gnavus*, have been shown to be associated with various intestinal and extraintestinal diseases and are increased in abundance in inflammatory bowel disease and metabolic diseases [30,31,32]. Unfortunately, limited by the accuracy of 16s analysis, the experiments were unable to explore the impact of *Salmonella* Typhimurium infection on duck intestinal microbiota at the species level.

## 5. Conclusions

This study demonstrates that *Salmonella* Typhimurium significantly alters the gut microbiota of ducks after infection and poses a serious public health risk. The core microbiomes altered in ducks after infection with *Salmonella* Typhimurium were mainly *Ruminococcaceae* and *Lachnospiraceae*. The potential pathways of the gut microbiome, such as succinate fermentation to butanoate (PWY-5677), superpathway of *Clostridium acetobutylicum* acidogenic fermentation (PWY-6590), L-1,2-propanediol degradation (PWY-7013), TCA cycle VIII (REDCITCYC), and other metabolic pathways, were significantly reduced after infection. In the future, we will explore more precise assays to explore new mechanisms for the prevention and control of *Salmonella* Typhimurium infections by screening potential key strains of the duck gut microbiome.

## Figures and Tables

**Figure 1 microorganisms-12-00602-f001:**
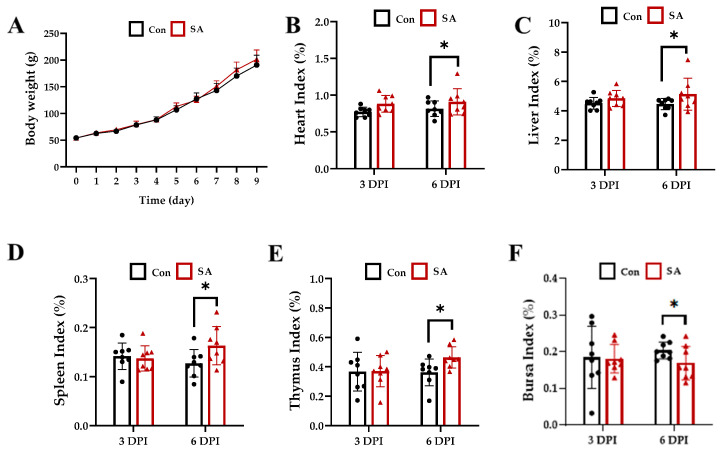
Effects of *Salmonella* Typhimurium infection on the body weight and organ development of ducks. (**A**) Effects of *Salmonella* Typhimurium infection on the body weight of ducks. (**B**–**F**) Effects of *Salmonella* Typhimurium infection on the heart, liver, spleen, thymus, and bursa index in ducks. Con: control group; SA: *Salmonella* Typhimurium group. The data are presented as the mean ± SD. *, *p* < 0.05.

**Figure 2 microorganisms-12-00602-f002:**
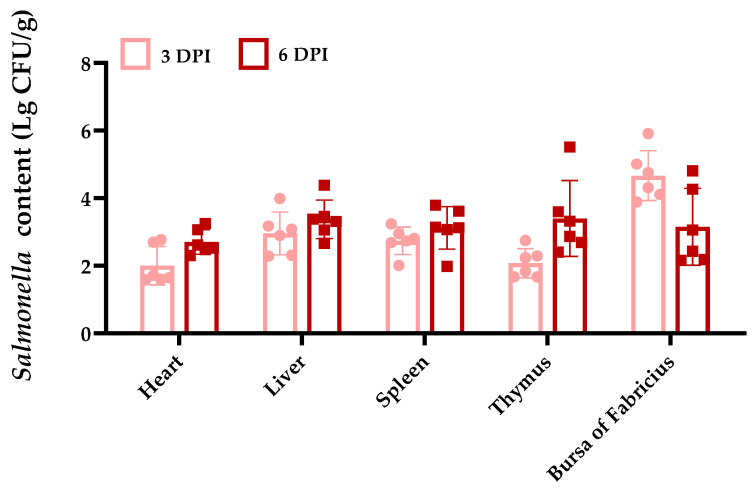
Colonization of ducks by *Salmonella* Typhimurium. The data are presented as the mean ± SD.

**Figure 3 microorganisms-12-00602-f003:**
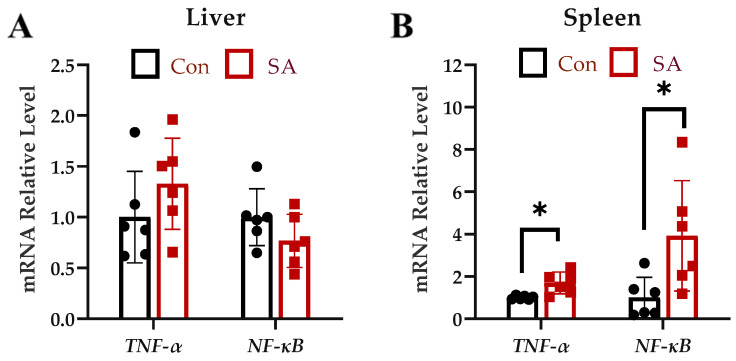
Effects of *Salmonella* Typhimurium infection on the expression of inflammatory cytokines on 6 dpi. (**A**) Liver mRNA expression of *TNF-α* and *NF-κB*. (**B**) Spleen mRNA expression of *TNF-α* and *NF-κB*. Con: control group; SA: *Salmonella* Typhimurium group. The data are presented as the mean ± SD. *, *p* < 0.05.

**Figure 4 microorganisms-12-00602-f004:**
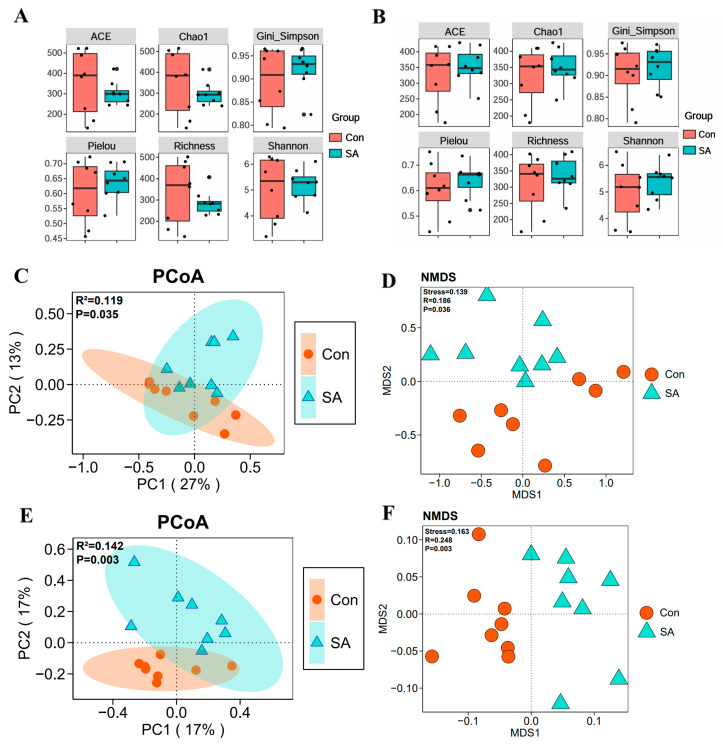
Diversity analysis of gut microbiota. (**A**) Alpha diversity (ACE, Chao1, Simpson, Pielou, Richness, and Shannon) at 3 dpi. (**B**) Alpha diversity at 6 dpi. (**C**) Principal coordinate analysis (PCoA) scores plot of the samples at 3 dpi. (**D**) PCoA at 6 dpi. (**E**) Nonmetric multidimensional scaling (NMDS) scores plot of the samples at 3 dpi. (**F**) NMDS at 6 dpi. Con: control group; SA: *Salmonella* Typhimurium group.

**Figure 5 microorganisms-12-00602-f005:**
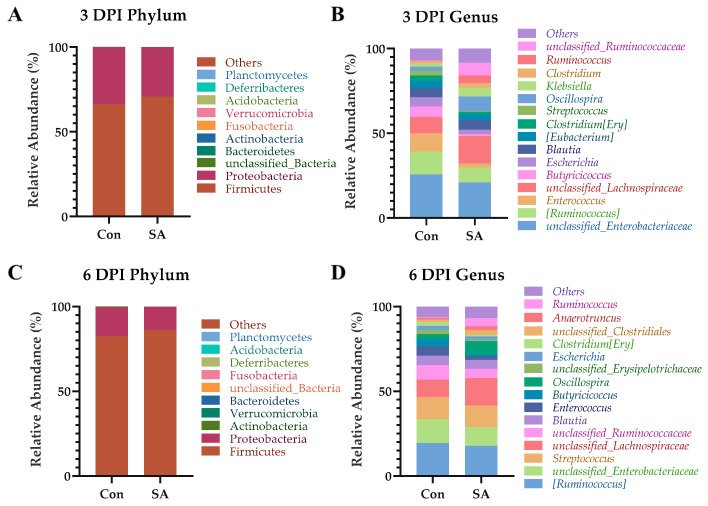
Effects of *Salmonella* Typhimurium on gut microbiota composition at the phylum and genus levels. (**A**) The top 10 phyla of gut microbiota in terms of abundance at 3 dpi. (**C**) Phyla at 6 dpi. (**B**) The top 15 genera of gut microbiota in terms of abundance at 3 dpi. (**D**) Genus at 6 dpi.

**Figure 6 microorganisms-12-00602-f006:**
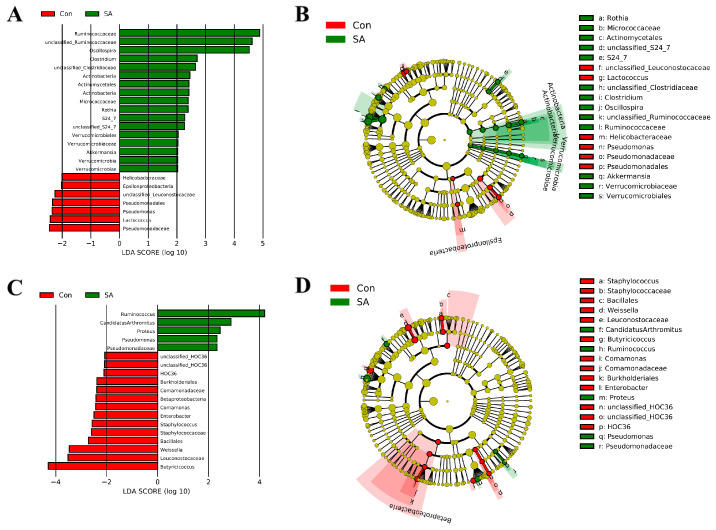
LEfSe analysis tree plots and histogram. (**A**) Histogram results at 3 dpi. (**B**) Taxonomic tree plots at 3 dpi. (**C**) Histogram results at 6 dpi. (**D**) Taxonomic tree plots at 6 dpi. Con: control group; SA: *Salmonella* Typhimurium group. Different colors indicate the enrichment of the biomarker taxa in Con (red) and SA (green) (LDA > 2). The histogram’s area represents the proportion of bacterial relative abundance. Con: control group; SA: *Salmonella* Typhimurium group.

**Figure 7 microorganisms-12-00602-f007:**
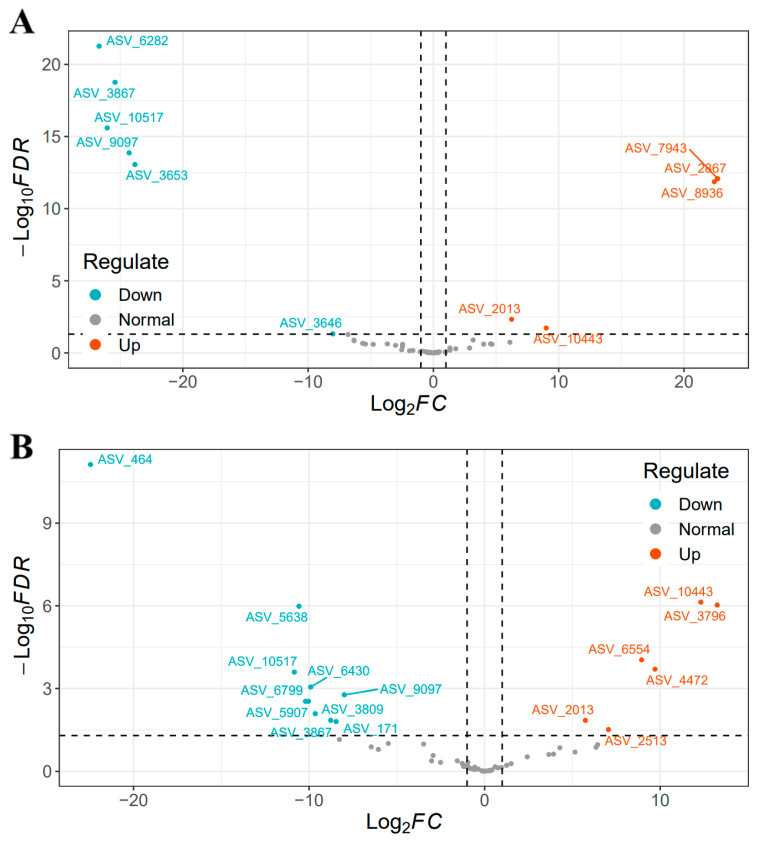
Volcano plot of differential strains. (**A**) On 3 dpi and (**B**) On 6 dpi. The red nodes represent significantly up-regulated strains in the control group, and the blue nodes represent significantly down-regulated strains (*p* < 0.05 and FC > 1). The grey nodes represent strains with no significance. Screening criteria: the total absolute abundance of the strain in the whole sample was greater than 1000 reads.

**Figure 8 microorganisms-12-00602-f008:**
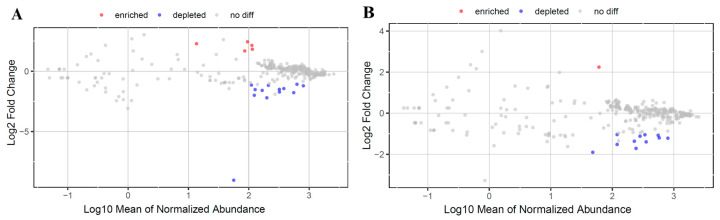
Volcano plot of differential microbe–metabolite pathways. (**A**) On 3 dpi and (**B**) 6 dpi. The red nodes represent significantly enriched strains in Con, while blue nodes represent significantly depleted strains (*p* < 0.05). The grey nodes represent strains with no significant differences.

**Table 1 microorganisms-12-00602-t001:** List and sequences of primers used in this study.

Gene	Genbank	Primer Position	Primer Sequences (5′→3′)
*TNF-α*	XM_046927265	Forward	CAGGACAGCCTATGCCAACA
		Reverse	ACAACCAGCTATGCACCCCA
*NF-κB*	XM_046915553	Forward	CAACGCAGGACCTAAAGACAT
		Reverse	CAGTAAACATAAGACGCACCACA
*β-actin*	L08165	Forward	GCTGTCCCTGTATGCCTCTG
		Reverse	TCTCGGCTGTGGTGGTGAAG

**Table 2 microorganisms-12-00602-t002:** Information list of differential strains within the volcano plot.

	Phylum	Class	Order	Family	Genus
ASV_9097	*Firmicutes*	*Clostridia*	*Clostridiales*	*Ruminococcaceae*	*Ruminococcus*
ASV_10517	*Firmicutes*	*Clostridia*	*Clostridiales*	*Ruminococcaceae*	*Ruminococcus*
ASV_6282	*Firmicutes*	*Clostridia*	*Clostridiales*	*Ruminococcaceae*	*unclassified_Ruminococcaceae*
ASV_3796	*Firmicutes*	*Clostridia*	*Clostridiales*	*Ruminococcaceae*	*unclassified_Ruminococcaceae*
ASV_464	*Firmicutes*	*Clostridia*	*Clostridiales*	*Ruminococcaceae*	*Oscillospira*
ASV_3867	*Firmicutes*	*Clostridia*	*Clostridiales*	*Lachnospiraceae*	*Blautia*
ASV_3653	*Firmicutes*	*Clostridia*	*Clostridiales*	*Lachnospiraceae*	*[Ruminococcus]*
ASV_7943	*Firmicutes*	*Clostridia*	*Clostridiales*	*Lachnospiraceae*	*unclassified_Lachnospiraceae*
ASV_5638	*Firmicutes*	*Clostridia*	*Clostridiales*	*Lachnospiraceae*	*unclassified_Lachnospiraceae*
ASV_6430	*Firmicutes*	*Clostridia*	*Clostridiales*	*Lachnospiraceae*	*unclassified_Lachnospiraceae*
ASV_10443	*Firmicutes*	*Clostridia*	*Clostridiales*	*Lachnospiraceae*	*[Ruminococcus]*
ASV_4472	*Firmicutes*	*Clostridia*	*Clostridiales*	*Lachnospiraceae*	*[Ruminococcus]*
ASV_6554	*Firmicutes*	*Bacilli*	*Lactobacillales*	*Leuconostocaceae*	*Weissella*
ASV_2867	*Firmicutes*	*Bacilli*	*Lactobacillales*	*Enterococcaceae*	*Enterococcus*
ASV_8936	*Firmicutes*	*Bacilli*	*Lactobacillales*	*Enterococcaceae*	*Enterococcus*

## Data Availability

The datasets produced and/or analyzed during the current study are available from the corresponding author upon reasonable request.

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
