# Peer review of "Effects of Salmonella Typhimurium Infection on the Gut Microbiota of Cherry Valley Meat Ducks"

_microorganisms, 2024, doi:10.3390/microorganisms12030602_

Round 1
Reviewer 1 Report
Comments and Suggestions for Authors
Thank you very much for the opportunity to participate in the review of the manuscript titled “Effects of Salmonella Typhimurium infection on the gut microbiota of Cherry Valley meat ducks”.
The authors of the manuscript showed that Salmonella Typhimurium can colonize ducks three days after infection and change the composition of the intestinal microflora, mainly by increasing the abundance of Ruminococcaceae and Lachnospiraceae. Salmonella Typhimurium infection significantly alters the intestinal microflora of ducks and poses a serious threat to public health.
Overall, the manuscript is very well and precisely written. I congratulate the authors for a great description. I have minor comments on the abstract (presented below). The introduction is correctly written and provided with appropriate citations to the literature. However, it should end with a clearly defined goal. The research material should be described in more detail. The methodology is written appropriately, although I lack references to the literature. The results are presented in an exemplary manner, and the description of the results is insightful and clear. The discussion refers to the presented results and is correct. The list of references could include more items from recent years.
I present my comments below.
Please correct keywords to words/expressions other than those in the title of the manuscript. This will increase the possibilities of searching for an article in databases.
Please correct the abstract so that the description includes the purpose of the research. It is also worth mentioning briefly the work methodology. What tests were performed?
Line 42. Please explain the GIT abbreviation.
Please add the purpose of the work to the introduction.
Where did the particular research methodologies come from? I miss the appropriate citations at the end of the subsections.
Line 63. Volume of what? What solution were the Salmonella bacteria in?
Line 64, 69... Please provide the manufacturers of reagents and microbiological media (producer, city, country).
Line 67. Please provide details about the Salmonella strain. Is this strain deposited in a Chinese collection? If so, please provide the number. Are the strain sequences deposited in the GenBank database? If so, please provide the number.
Line 84. What conditions were the plates incubated?
Line 91. Table 1 instead of Table 2.
Figure 1. The axes on the charts should be labeled more precisely. For example, time (days) or body weight (g). Please correct all charts.
Figure 2 is missing. Please complete.
Figures 4A and 4B are blurred. Please enlarge the figures to make them more legible. Please complete the units.
Line 240. Please write Clostridium acetobutylicum in italics.
Discussion. Please correct the names of microorganisms to italics.
I would like to ask the authors to comment on how the presented results can be used in poultry breeding in the future? Is there any application potential here?
Author Response
Thank you very much for taking the time to review this manuscript. Please find the detailed responses below and the corresponding revisions/corrections highlighted/in track changes in the re-submitted files. If you have any questions, please let me know.
Comments 1: Please correct keywords to words/expressions other than those in the title of the manuscript. This will increase the possibilities of searching for an article in databases.
Response 1: We greatly thank the comment. Thank you very much for your suggestion. We have changed the keyword to "Salmonella Typhimurium, gut microbiota dysbiosis, ducks, Ruminococcaceae, 16S rRNA". The relevant content has been modified in the text. (Line 29)
Comments 2: Please correct the abstract so that the description includes the purpose of the research. It is also worth mentioning briefly the work methodology. What tests were performed?
Response 2: The relevant content has been modified in the article. Supplemented the research purpose "This study uses Salmonella typhimurium as the research object and Cherry Valley meat duck as the model to study the impact of Salmonella infection on ducks.", and related working methods "After infection, duck viscera were collected to detect the colonization of Salmonella, and cecal contents were collected to analyze the changes in gut microbiota." (Line 20-28)
Comments 3: Line 42. Please explain the GIT abbreviation.
Response 3: We greatly thank the comment. We apologize for the mistake due to our negligence. GIT: gastrointestinal tract. The relevant content has been modified in the article. (Line 48)
Comments 4: Please add the purpose of the work to the introduction.
Response 4: We greatly thank the comment. The relevant content has been modified in the article. (Line 20-22)
Comments 5: Where did the particular research methodologies come from? I miss the appropriate citations at the end of the subsections.
Response 5: We greatly thank the comment. Thank you for pointing this out. According to the literature in the field of microbiome in recent years, the methods for sequencing region and primer selection have been well developed. Methods were standardized across institutions. In general, for the sake of article simplicity, such generic methods are not referenced.
Comments 6: Line 63. Volume of what? What solution were the Salmonella bacteria in?
Response 6: We greatly thank the comment. Before infection, single colonies were picked and cultured in 5mL Lysogeny broth with shaking for 16 hours. The bacteria were collected by centrifugation, resuspended in physiological saline, counted, and the Salmonella was mixed into 4*108CFU/mL using physiological saline, and each duck was administered 0.5 mL. Relevant content has been added to the article. (Line81-87)
Comments 7: Line 64, 69... Please provide the manufacturers of reagents and microbiological media (producer, city, country).
Response 7: We greatly thank the comment. Relevant content has been added to the article. (Line 81-87)
Comments 8: Line 67. Please provide details about the Salmonella strain. Is this strain deposited in a Chinese collection? If so, please provide the number. Are the strain sequences deposited in the GenBank database? If so, please provide the number.
Response 8: We greatly thank the comment. Salmonella Typhimurium 20JS04 strain was isolated from sick ducks in a Cherry Valley meat duck factory in Shandong Province, China in 2022. The strain was not deposited in the Chinese Collection or uploaded to the GenBank database. Relevant content has been added to the article. (Line 78-80)
Comments 9: Line 84. What conditions were the plates incubated?
Response 9: We greatly thank the comment. Bacteria were counted after the culture plate was incubated in a 37°C incubator for 16 h. Relevant content has been added to the article. (Line 101)
Comments 10: Line 91. Table 1 instead of Table 2.
Response 10: We greatly thank the comment. We apologize for the mistake due to our negligence. The relevant content has been modified in the article. (Line 109)
Comments 11: Figure 1. The axes on the charts should be labeled more precisely. For example, time (days) or body weight (g). Please correct all charts.
Response 11: We greatly thank the comment. The relevant content has been modified in the article. (Line 156)
Comments 12: Figure 2 is missing. Please complete.
Response 12: We greatly thank the comment. The relevant content has been modified in the article. The image has been re-uploaded, possibly due to formatting issues. (Line 171)
Comments 13: Figures 4A and 4B are blurred. Please enlarge the figures to make them more legible. Please complete the units.
Response 13: We greatly thank the comment. The relevant content has been modified in the article. (Line 198)
Comments 14: Line 240. Please write Clostridium acetobutylicum in italics.
Response 14: We greatly thank the comment. We apologize for the mistake due to our negligence. The relevant content has been modified in the article. (Line 262)
Comments 15: Discussion. Please correct the names of microorganisms to italics.
Response 15: We greatly thank the comment. We apologize for the mistake due to our negligence. We will check the full text and make revisions.
Comments 16: I would like to ask the authors to comment on how the presented results can be used in poultry breeding in the future? Is there any application potential here?
Response 16: We greatly thank the comment. Salmonella infection seriously harms the duck breeding industry and poses a public health threat. Salmonella resistance is a serious problem due to the overuse of antibiotics, and microbial therapies may be the key to solving this problem. We are trying to find a type of microorganisms that can prevent or treat Salmonella infection by analyzing the changes in gut microbiota It may be possible to establish a microbial protective barrier during the early stages of duck rearing by directly supplementing the microorganisms or by providing substances that influence these microorganisms after Salmonella infection.

Reviewer 2 Report
Comments and Suggestions for Authors
There are numerous studies to understand Salmonella infection in pigs. This unique research seeks to establish the relationship between Salmonella and the intestinal microbiota in ducks and attempts to establish some controllable parameters that can eradicate this type of infection in the first stage of development. For this, appropriate and comparative analysis techniques are used and the results are interpreted analytically and statistically. The obtained scientific results represent an important step and provide a valuable theoretical basis in the fight for the prevention and treatment of infections caused by Salmonella bacteria in ducks.
Author Response
Thank you very much for taking the time to review this manuscript. Please find the detailed responses below and the corresponding revisions/corrections highlighted/in track changes in the re-submitted files. If you have any questions, please let me know.
Comments 1: There are numerous studies to understand Salmonella infection in pigs. This unique research seeks to establish the relationship between Salmonella and the intestinal microbiota in ducks and attempts to establish some controllable parameters that can eradicate this type of infection in the first stage of development. For this, appropriate and comparative analysis techniques are used and the results are interpreted analytically and statistically. The obtained scientific results represent an important step and provide a valuable theoretical basis in the fight for the prevention and treatment of infections caused by Salmonella bacteria in ducks.
Response 1: Thank you very much for liking this article. Your comments will be of great help to our article and future research direction. Much remains unknown in the field of duck gut microbiome research. We will work harder in the future to find more important and valuable results.

Reviewer 3 Report
Comments and Suggestions for Authors
1. Lines 49-52: It is not clear to me what the authors' objective is. In the abstract they talk about the importance of the pathogen in public health, in the introduction they explain in good detail the pathogenesis mediated by the pathogen, the importance or impact it has on the health of ducks and the farms that raise these animals. However, they do not make clear the main objective and impact. I am not saying that the article is not relevant, only that the wording is not textually clear, in my opinion. Additionally, the introduction does not include much about the impact of the pathogen on the microbiota or the microbiota and its importance.
2. Lines 58-60: Why do the authors use this particular animal model?
3. Why did they use Salmonella Typhimurium strain 20JS04 in particular, and not an ATCC strain. Is this strain important for the authors?
4.- Lines 71-79: Why, if the article focuses on the changes generated by salmonella in the intestinal microbiota, did the authors use other organs of the duck? I think this should be made clear in the introduction.
5. Table 1 has no title. It is also not mentioned in the methodology text because these primers were used. I understand that they are the controls, but it should be indicated in the text.
6. Lines 100-121: Why these primers, what do they code for, what is the importance of the V3-V4 regions. I believe that this information should be included in this section. In addition, if the primers are of own design it should be indicated and if they come from another publication it should be cited.
7. In Figure 1. What represents each point of the bar graph. I understand that they are determining the change in weight of each organ they extracted (that answers one of my previous doubts). There were 3 experiments performed for each assay, so I assume that each point on the graphs represents a duck. However, in the methodology they mention that they selected 6 ducks and in the bars I see 8 points. If I am confused, please clarify. On the other hand, in this same figure, I see that the SD at 6 DPI is high in the group with SA and I wonder if the statistical significance is not influenced by that SD, could the authors corroborate this by eliminating the data out of range and perform the analysis again? It could also show the results for each duck separately. To see if the change is really statistically significant.
8. Lines 144-149: I am not very sure, I consider that the out-of-range data should be eliminated to reduce the SD and again perform the statistical analysis. If they were different isolates it could be explained, but they are the same strain. It causes me doubt, please check it. It could also show the results for each duck separately. To see if the change is really statistically significant. I find it strange the difference so marked per animal, considering that theoretically they should be in the same conditions pre and post experiment.
9.- Figure 2 is not in the text.
10. Lines 154-158: It is not clear to me the relationship between the pathogen and the expression of these cytosines. I know that they are associated with inflammation, but it is not clear in the text the purpose of implementing them. I feel that the experimental design is not well supported in the text.
11. Lines 164-175: Shouldn't this result come after showing the microbiota of ducks with and without salmonella?
12. Creo que es prudente que muestren los resultados para cada animal por separado. Incluyendo aquellos en los que se determino la composición de la microbiota con y sin salmonella. Me parece extraño que haya tanta variación si los patos estaban bajo las mismas condiciones y las DS de sus experimentos, algunos casos , muestran valores muy altos. No creo que sea prudente concluir hasta realizar este analisis.
13. Table two has no title.
14. It is important that the authors analyze separately the result for each duck before concluding because I observed important variations per study subject that theoretically should not be observed if they were the same conditions pre and post experiment.
Best wishes.
Author Response
Thank you very much for taking the time to review this manuscript. Please find the detailed responses below and the corresponding revisions/corrections highlighted/in track changes in the re-submitted files. If you have any questions, please let me know.
|
Comments 1: Lines 49-52: It is not clear to me what the authors' objective is. In the abstract they talk about the importance of the pathogen in public health, in the introduction they explain in good detail the pathogenesis mediated by the pathogen, the importance or impact it has on the health of ducks and the farms that raise these animals. However, they do not make clear the main objective and impact. I am not saying that the article is not relevant, only that the wording is not textually clear, in my opinion. Additionally, the introduction does not include much about the impact of the pathogen on the microbiota or the microbiota and its importance. Response 1: We greatly thank the comment. We have revised and added to the introduction. (Line 44-61) Comments 2: Lines 58-60: Why do the authors use this particular animal model? Response 2: We greatly thank the comment. Cherry Valley duck is one of the most famous duck breeds in China, and this strain of Salmonella was also isolated from Cherry Valley duck. Comments 3: Why did they use Salmonella Typhimurium strain 20JS04 in particular, and not an ATCC strain. Is this strain important for the authors? Response 3: We greatly thank the comment. Salmonella Typhimurium 20JS04 strain was isolated from sick ducks in a Cherry Valley meat duck factory in Shandong Province, China in 2022. We believe that the results obtained by using Salmonella isolated from actual sick ducks can be more applied to actual production. Comments 4: Lines 71-79: Why, if the article focuses on the changes generated by salmonella in the intestinal microbiota, did the authors use other organs of the duck? I think this should be made clear in the introduction. Response 4: We greatly thank the comment. We sought to confirm the success of the model and the public health threat of this strain of Salmonella by detecting the amount of Salmonella colonization in the organs. Comments 5: Table 1 has no title. It is also not mentioned in the methodology text because these primers were used. I understand that they are the controls, but it should be indicated in the text. Response 5: We greatly thank the comment. We apologize for the mistake due to our negligence. The relevant content has been modified in the article. (Line 109, 116) Comments 6: Lines 100-121: Why these primers, what do they code for, what is the importance of the V3-V4 regions. I believe that this information should be included in this section. In addition, if the primers are of own design it should be indicated and if they come from another publication it should be cited. Response 6: We greatly thank the comment. According to the literature in the field of microbiome in recent years, the methods for sequencing region and primer selection have been well developed. Methods were standardized across institutions. In general, for the sake of article simplicity, such generic methods are not referenced. Each strain has a unique genetic sequence that distinguishes it from others. In the whole gene sequence, the V3-V4 region is a specific region, which is selected as the method to distinguish strains in the most 16S rRNA sequencing. The primers don’t have code, there sequences are already unique. The experiment primers were not designed by ourselves, but also used in the vast majority of other experiments. This generic method is usually not referenced in bioinformatic analysis researches. Comments 7: In Figure 1. What represents each point of the bar graph. I understand that they are determining the change in weight of each organ they extracted (that answers one of my previous doubts). There were 3 experiments performed for each assay, so I assume that each point on the graphs represents a duck. However, in the methodology they mention that they selected 6 ducks and in the bars I see 8 points. If I am confused, please clarify. On the other hand, in this same figure, I see that the SD at 6 DPI is high in the group with SA and I wonder if the statistical significance is not influenced by that SD, could the authors corroborate this by eliminating the data out of range and perform the analysis again? It could also show the results for each duck separately. To see if the change is really statistically significant. Response 7: We greatly thank the comment. We collected the samples only at 3 dpi and 6 dpi, which means Day 6 and Day 9. On both time nodes, we selected 8 ducks per group (Con and SA), and they all collected the organ index. That is why there were 8 points (Figure 1). However, only 6 of the duck organ sample to detect the number of Salmonella bacteria and quantitative real-time PCR (QPCR). We mentioned in the Line 73-79, and that is why there were 6 points in Figure 3. For the outlier samples, all the data in this experiment were true and credible, and any outlier data were recorded trutily. The statistical results are also real and in line with the experimental expectation. Comments 8: Lines 144-149: I am not very sure, I consider that the out-of-range data should be eliminated to reduce the SD and again perform the statistical analysis. If they were different isolates it could be explained, but they are the same strain. It causes me doubt, please check it. It could also show the results for each duck separately. To see if the change is really statistically significant. I find it strange the difference so marked per animal, considering that theoretically they should be in the same conditions pre and post experiment. Response 8: We greatly thank the comment. Even if each sample uses the same strain and the same treatment, there will be gaps between individuals, and it is impossible for every data to be completely consistent. All the data in the experiment were within the error range (P value), and the statistical results were true and credible. We only have to show that there is a significant difference between the CON and SA groups in the overall range to satisfy the experimental purpose. Comments 9: Figure 2 is not in the text. Response 9: We greatly thank the comment. The relevant content has been modified in the article. The image has been re-uploaded, possibly due to formatting issues. (Line 161) Comments 10: Lines 154-158: It is not clear to me the relationship between the pathogen and the expression of these cytosines. I know that they are associated with inflammation, but it is not clear in the text the purpose of implementing them. I feel that the experimental design is not well supported in the text. Response 10: We greatly thank the comment. Salmonella infection can cause intestinal inflammation, creating an intestinal environment suitable for the proliferation and spread of Salmonella. The purpose of detecting the expression of these cytokines is to confirm the impact of Salmonella on meat ducks and to successfully construct the experimental model. Comments 11: Lines 164-175: Shouldn't this result come after showing the microbiota of ducks with and without salmonella? Response 11: We greatly thank the comment. Before lines 164, all samples were analyzed for physiological and biochemical indicators. The subsequent results were based on the data analysis of the gut microbiota of the samples. For the Figure 4, on 3 dpi (Day 6), we collected 8 ducks per group (Con and SA). And on 6 dpi (Day 9), we collected another 8 ducks per group. Comments 12: Creo que es prudente que muestren los resultados para cada animal por separado. Incluyendo aquellos en los que se determino la composición de la microbiota con y sin salmonella. Me parece extraño que haya tanta variación si los patos estaban bajo las mismas condiciones y las DS de sus experimentos, algunos casos , muestran valores muy altos. No creo que sea prudente concluir hasta realizar este analisis. Response 12: We greatly thank the comment. Gut microbiota, mainly composed of bacteria and fungi, are numerous and closely affect each other. Just slight variations can cause differences between individual samples. Therefore, ducks within the same treatment group may also have large differences. To this end, we use as many samples as possible for comprehensive analysis. The data of each sample in the experiment is real and credible and should not be discarded. Comments 13: Table two has no title. Response 13: We greatly thank the comment. We greatly thank the comment. We apologize for the mistake due to our negligence. The relevant content has been modified in the article. (Line 252) Comments 14: It is important that the authors analyze separately the result for each duck before concluding because I observed important variations per study subject that theoretically should not be observed if they were the same conditions pre and post experiment. Response 14: We greatly thank the comment. When testing with live animals, it’s difficult to ensure that every animal is the same, especially when it comes to gut microbiota. To this end, we use as many samples as possible for comprehensive analysis. The data of each sample in the experiment is real and credible and should not be discarded. |
